# Association of Dietary Patterns with Metabolic Syndrome in Chinese Children and Adolescents Aged 7–17: The China National Nutrition and Health Surveillance of Children and Lactating Mothers in 2016–2017

**DOI:** 10.3390/nu14173524

**Published:** 2022-08-26

**Authors:** Jia Shi, Hongyun Fang, Qiya Guo, Dongmei Yu, Lahong Ju, Xue Cheng, Wei Piao, Xiaoli Xu, Zizi Li, Di Mu, Liyun Zhao, Li He

**Affiliations:** National Institute for Nutrition and Health, Chinese Center for Disease Control and Prevention, Beijing 100050, China

**Keywords:** dietary patterns, metabolic syndrome, children and adolescents, China

## Abstract

This study aims to determine the associations of dietary patterns with metabolic syndrome (MetS) and its components in Chinese children and adolescents aged 7–17 in 2016–2017. Using the data from the China National Nutrition and Health Surveillance of Children and Lactating Mothers in 2016–2017, the sociodemographic information, diet, anthropometric measurements and clinical examinations of subjects were obtained, and a total of 13,071 school-aged children and adolescents were included in this study. The Cook criteria were used to define MetS and its components. Dietary intake was derived from 24-h dietary records for three consecutive days, combined with the weighing method. Factor analysis was used to identify major dietary patterns. The associations of dietary patterns with MetS and its components were examined by logistic regression analysis. Consequently, five distinct dietary patterns were identified by factor analysis, and the relationships between dietary patterns with MetS and its components were observed. After adjusting for covariates, the animal product and vegetable patterns may have a positive association with MetS; the condiment pattern was positively associated with low HDL-C; the fruit and junk food patterns had positive relationships with MetS, abdominal obesity and high TG; the cereals and tubers pattern was positively associated with MetS, abdominal obesity, high TG and low HDL-C; the beans pattern was positively associated with high TG.

## 1. Introduction

Metabolic syndrome (MetS), also known as “Syndrome X” and “Insulin Resistance Syndrome”, is a complex group of metabolic disorders of energy use and storage, characterized by the combination of abdominal obesity, elevated blood pressure (BP), high triglycerides (TG), elevated fast blood glucose (FBG) and low high-density lipoprotein cholesterol (HDL-C) [1]. This syndrome is one of the most serious non-communicable chronic diseases and it increases the risk of type 2 diabetes and cardiovascular disease (CVD) [2]. However, the prevalence of metabolic syndrome increases every year worldwide. A meta-analysis of MetS prevalence in Chinese children revealed that the prevalence of MetS diagnosed by Cook’s criteria increased from 2.3% in 2004–2010 to 3.2% in 2011–2014 [3]. The dramatic increasing trend may be associated with the epidemic of obesity in China [4]. Although the prevalence is lower than that in developed countries, there are at least 11 million children in China affected by this syndrome because of the huge population base [5,6,7].

Important factors associated with MetS include genetic, socio-economic and environmental factors, unhealthy dietary habits and urbanization development [8]. However, diet, as an important part of lifestyle, has been suggested to be significantly associated with metabolic syndrome [9,10]. Multiple studies have indicated that the Mediterranean diet [11] and the DASH diet [9] have positive effects on MetS. Moreover, diet is one of the most modifiable and important risk factors in the etiology of chronic diseases; the identification of eating patterns characterized by the consumption of “unhealthy” foods in childhood and adolescence can be useful for the development of strategies that improve dietary habits and reduce the prevalence of these risk factors throughout life [12]. Dietary pattern analysis has been suggested as a useful measure for understanding the relationship between the overall quality of diets and health outcomes [13]. The result of diet pattern analysis might be best explained by the focus on the overall diet rather than the health effects of a single food or nutrient [14]. Therefore, studies involving dietary patterns and their associations with metabolic syndrome should be performed in children and adolescents, aiming at interventions and early changes in dietary habits. However, these studies are not very common in children and adolescents and more information is still necessary [15].

Thus, this study is designed to investigate whether dietary patterns and MetS are correlated in a national sample of 7–17-year-old Chinese children and adolescents.

## 2. Materials and Methods

### 2.1. Study Design and Participants

Data were obtained from the China National Nutrition and Health Surveillance of Children and Lactating Mothers in 2016–2017. A complex, stratified, multi-stage cluster random sampling design was performed to provide a representative sample of 31 provinces, autonomous regions and municipalities in China. A total of 275 survey sites were randomly selected. Two townships/subdistricts were randomly chosen from every survey site, and two communities/villages were randomly selected from each township/subdistrict. At least 280 children and adolescents aged 6–17 were selected in school from each survey site, with equal numbers of males and females. The details of the study design have been explained in an earlier publication [16]. Participants with abnormal dietary energy intake (<600 or >4000 kcal/day for subjects aged 7–11; <800 or >4500 kcal/day for subjects aged 12–17) or with missing data on sociodemographic information, anthropometric measurements and clinical examinations were excluded. Consequently, data of 13,071 participants aged 7–17 were included in the analysis.

### 2.2. Ethics Approval and Consent to Participate

This study was approved by the ethical committee of the National Institute for Nutrition and Health of the Chinese Center for Disease Control and Prevention. The ethical approval number was 201614. All participants provided informed consent before the study, either by themselves or via their guardian.

### 2.3. Metabolic Syndrome

Cook’s criteria [17], modified for age on the basis of the National Cholesterol Education Program—Adult Treatment Panel III (NCEP-ATP III), were chosen to define MetS. According to Cook’s criteria, participants meeting more than 3 of the following 5 criteria were defined as having the MetS:(1)Abdominal obesity: WC ≥ age- and sex-specific 90th percentile, determined by the cutoff points for Chinese children and adolescents [18];(2)Elevated blood pressure: SBP or DBP ≥ 90th percentile for age, sex and height [19];(3)High triglycerides: serum TG ≥ 1.24 mmol/L;(4)Low HDL-C: HDL-C ≤ 1.03 mmol/L;(5)Elevated fast blood glucose: FBG ≥ 6.1 mmol/L.

### 2.4. Anthropometric Measurements and Clinical Examinations

The measurements were conducted by trained investigators. Anthropometric measurements included height, waist circumference (WC), systolic blood pressure (SBP), diastolic blood pressure (DBP). The height was measured to the nearest 0.1 cm with a stadiometer (TZG) with the participants standing in an upright position. The WC measurement was taken twice with a soft tape at the midpoint between the bottom of the rib cage and above the top of the iliac bone. We ensured that the error between the two measurements was less than 2 cm and recorded the average value. After resting for at least 5 min, SBP and DBP were measured three times in one-minute intervals by an electronic sphygmomanometer (Omron HBP 1300, Tokyo, Japan) with accuracy to 1 mmHg, and the mean of SBP and DBP were calculated and considered as an individual’s blood pressure.

Clinical examinations included the concentration of triglycerides (TG), high-density lipoprotein cholesterol (HDL-C) and fast blood glucose (FBG). In this study, 6 mL fasting venous blood samples of participants were obtained in the morning and used for clinical testing. TG and HDL-C were tested using enzyme colorimetry (Roche Cobas C701 automatic biochemical analyzer series). FBG was measured using the glucokinase method (Roche P800 automatic biochemical analyzer).

### 2.5. Assessment of Other Variables

Data on sociodemographic variables such as sex, age, living area, income, food expenses, physical activity, smoking and alcohol drinking were collected by a standardized questionnaire provided by the China Center for Disease Control and Prevention (the China CDC) project group.

The living area was dichotomized into urban and rural areas. In this study, we divided all the subjects into three age groups (7–10, 11–13 and 14–17 for females; 7–11, 12–14 and 15–17 for males) to reflect the prepubertal, pubertal and post-pubertal stages, respectively, according to the Chinese classification [7]. Engel’s coefficient was the proportion of total food expenditure to total personal consumption expenditure and was divided into 6 categories in this study (≥60%, 50–59%, 40–49%, 30–39%, <30% and unknown, respectively). The higher the Engel’s coefficient, the worse the family economic condition. Physical activity was defined as low level (0–3 days/week), high level (≥4 days/week) and unknown, according to a previous study [20]. The smoking status was divided into five groups (everyday, 4–6 days/week, 1–3 days/week, <1 day/week and none). Alcohol drinking status was divided into three groups (drinking within 30 days, 30 days ago and never).

### 2.6. Dietary Assessment

All dietary information was collected using the 24-h dietary record method for three consecutive days (two weekdays and one weekend day) by professionally trained investigators. Condiment consumption was also recorded using the weighing method. The average amount of each type of food group and energy intake was calculated by each individual (grams per day per reference man) for data analysis. Dietary energy intake was calculated based on the Chinese Food Composition Tables [21,22].

### 2.7. Dietary Pattern Analysis

For dietary pattern analysis, food variables were generated first because quantitative data such as the 3 days of food consumption were used. A total of 14 food groups based on similarity and food group classification were finally included for factor analysis with varimax rotation to explore the latent dietary pattern (DP) in our samples, including cereals (mainly rice and wheat products), tubers (mainly potatoes and sweet potatoes), mixed beans (such as kidney beans, mung beans, adzuki beans), legumes (mainly soybeans), vegetables, fruits, nuts (such as almonds, walnuts, peanuts and seeds), meat and poultry (mainly pork, beef, lamb, chicken, duck and processed meat), fish and shrimp, milk, eggs, junk food (fast foods, ethnic foods and cakes), oil (vegetable oil and animal oil) and salt. The number of factors was determined based on eigenvalues (eigenvalue > 1), the scree plot and the interpretability of the derived factors [13]. Each DP was named by the characteristics of food variables whose absolute factor loading was over 0.20 [23]. The food variables with higher factor loadings within DPs indicated higher consumption. Every participant had a number of factor scores and the largest score indicated that their diet was inclined towards the corresponding DP. All subjects were divided into four groups according to the quartile of each DP score.

### 2.8. Statistical Analysis

All analyses were conducted with SAS software (v.9.4, SAS Institute Inc., Cary, NC, USA) and the plot in this study was created using R software (Version 4.1.2). The categorical data were reported as numbers (percentages) and the Rao–Scott Chi-square test was performed to compare the distribution of different characteristics between subgroups. In order to maintain national representativeness, the PROC SURVEYFREQ program was applied. The weight of the sample was calculated by data from the China National Bureau Statistics in 2010. Factor analysis was conducted by the PROC FACTOR program. Multivariate logistic regression analysis was applied to explore the associations between DPs with MetS and its components. The model was adjusted for sex, residence area, age, Engel’s coefficient, physical activity, smoking and alcohol drinking, and all the covariables were transformed into dummy variables before conducting the adjustive process. The odds ratios (ORs), 95% confidence intervals (CIs) and *p* for trends were presented. Two-sided *p* values < 0.05 were considered to be statistically significant.

## 3. Results

### 3.1. Characteristics of Participants

Characteristics of 13,071 children and adolescents aged 7–17 are displayed in Table 1. In this sample, the prevalence of MetS was 5.36%. Among the participants, 53.48% were males and 46.62% were females. Male had a higher prevalence of MetS than females (male: 3.14% vs. female: 2.22%, *p* < 0.05). Moreover, 46.58% of all subjects were living in urban areas and 53.42% were living in rural areas. Participants classified as “living in urban areas” showed a higher prevalence of MetS than those living in rural areas (urban: 2.93% vs. rural: 2.42%, *p* < 0.0001). MetS was more prevalent in participants at post-pubertal stage (post-pubertal: 2.32% vs. pubertal: 1.51% vs. prepubertal: 1.53%, *p* < 0.0001). The rate of participants who engaged in physical activity was at least 78.12%. However, it was noted that the number of participants who were smoking was up to 45.03% and the drinking rate was 16.94%.

### 3.2. Dietary Patterns and Its Distribution among Children and Adolescents Aged 7–17

As shown in Figure 1, five mutually exclusive dietary patterns were identified by factor analysis, which together accounted for 45.311% of the total variation. Five DPs could explain 10.58%, 9.31%, 9.18%, 8.75% and 7.49% of the variance, respectively. DP1 was named “animal product and vegetable pattern”, characterized by high factor loadings from meat and poultry, fish and shrimp, vegetables, eggs and legumes. Similarly, the other four DPs were named DP2—condiments pattern (rich in oils and salt), DP3—fruit and junk food pattern (rich in fruits, junk food, nuts, eggs and tubers), DP4—cereals and tubers pattern (rich in cereals, tubers and vegetables and low in junk food) and DP5—beans pattern (rich in legumes, mixed beans and tubers, but low in eggs).

The results of the distribution of five DPs are presented in Table 2. Among children and adolescents aged 7–17, the percentages of DP1 to DP5 were 19.24%, 16.32%, 19.39%, 24.38% and 20.66%, respectively. Obviously, DP4 (cereals and tubers pattern) was the main DP in all subjects. However, DP1 (animal product and vegetable pattern) appeared in the participants living in urban areas. DP5 (beans pattern) appeared in the participants whose Engel’s coefficient was greater than 40% and participants who were drinking.

### 3.3. Associations between Dietary Patterns with MetS and Its Components

The results of associations between DPs with MetS and its components are shown in Table 3. Compared with Q1, participants in the Q2 level of DP1 had a positive influence on metabolic syndrome after adjusting for covariates (OR = 1.256, 95%CI: 1.003–1.572), and the DP1 score was negatively associated with low HDL-C (*p* for trend < 0.0001). DP2 was significantly associated with decreased odds of abdominal obesity (*p* for trend < 0.05) and elevated FBG (OR = 0.897, 95%CI: 0.810–0.992), and increased odds of low-HDL-C (Q4 vs. Q1, OR = 1.172, 95%CI: 1.002–1.370). DP3 was positively associated with metabolic syndrome (Q4 vs. Q1, OR = 1.367, 95%CI: 1.079–1.732, *p* for trend < 0.05) and abdominal obesity (*p* for trend < 0.0001), high TG (Q4 vs. Q1, OR = 1.156, 95%CI: 1.003–1.332, *p* for trend < 0.05), but was negatively associated with low HDL-C (Q4 vs. Q1, OR = 0.833, 95%CI: 0.700–0.991). Positive associations were observed between DP4 and metabolic syndrome (Q4 vs. Q1, OR = 1.315, 95%CI: 1.057–1.636), abdominal obesity (Q4 vs. Q1, OR = 1.357, 95%CI: 1.188–1.551, *p* for trend < 0.05), high TG (Q4 vs. Q1, OR = 1.165, 95%CI: 1.019–1.332) and low HDL-C (*p* for trend < 0.05). DP5 was negatively associated with abdominal obesity (Q4 vs. Q1, OR = 0.862, 95%CI: 0.751–0.989) and positively associated with high TG (Q3 vs. Q1, OR = 1.182, 95%CI: 1.029–1.357).

## 4. Discussion

In the present study, five dietary patterns among Chinese children and adolescents aged 7–17 were identified using factor analysis, including the animal product and vegetable pattern, condiment pattern, fruit and junk food pattern, cereals and tubers pattern and beans pattern. The relationships between DPs and MetS and its components were observed.

Among the five DPs, DP4—the cereals and tubers pattern—was mainly observed in all the subjects, followed by the beans pattern and fruit and junk food pattern, which indicated that there was a predominantly plant-based diet among subjects. DP1 (animal product and vegetable pattern) appeared in the participants living in urban areas, where there is an abundant supply of food and better economic conditions. Additionally, DP5 (beans pattern) appeared in the participants whose Engel’s coefficient was greater than 40% and participants who drank alcohol. In other words, DP5 appeared in the participants who had a worse economic condition and an unhealthy lifestyle. Meanwhile, it was also noted that the rate of participants who were smoking was up to 45.03% and the drinking rate was 16.94%, and immediate intervention is necessary.

DP1, named the animal product and vegetable pattern in our study, which was rich in meat and poultry, fish and shrimp, vegetables, eggs and legumes, was found to have a positive association with metabolic syndrome. This dietary pattern was similar to the modern dietary pattern of nutrition and diet investigation projects in Jiangsu Province of China [24] and the China Health and Nutrition Survey (CHNS) [25,26], which was high in content of pork, vegetables, seafood, pastry food and other animal meats. This dietary pattern, containing high amounts of animal meats (including processed meat and red meat), is closer to the Western dietary pattern [27], which is rich in saturated fats [28,29]. A previous prospective study of young adults had suggested that greater adherence to unhealthy Western diet patterns predicted a higher risk of MetS and increased insulin resistance [30]. A cohort study also indicated that the Western dietary pattern was negatively associated with HDL-C [10]. However, a negative association between DP1 and low HDL-C was observed in our study, and the *p* for this trend was statistically significant. Given the low intake of fish and shrimp and nuts, we speculated that the vegetables and legumes in DP1 may provide protection against low HDL-C. A few studies had reported that the consumption of dietary fiber contained in vegetables could improve lipid profiles and have beneficial effects on HDL-C [31,32,33]. In addition, a study also suggested that even Asians who consume substantial meat or animal-based foods still adhere to a cereal-based diet, which could contribute to the maintenance of a relatively low-fat diet, compared with Western countries [34].

DP2, named the condiment pattern, was characterized by high factor loadings from oil and salt. To our surprise, DP2 was found to be negatively associated with abdominal obesity and elevated BP. In general, salty food could stimulate the brain to crave them and also be responsible for the increased incidence of obesity [35]. Furthermore, high salt intake was considered as an independent risk factor for hypertension [36,37]. After further analysis of the raw data, we hypothesized that the reason for this result might be that the main source of oil consumed by children and adolescents was vegetable oil. Compared with animal oil, vegetable oil contains large amounts of unsaturated fatty acids, especially omega-3 fatty acids, which have been reported to improve lipocalin concentrations, insulin resistance and body composition [38,39,40]. Omega-3 fatty acid intake is associated with low levels of blood pressure and high levels of HDL-C, WC and obesity prevalence [41]. However, it has also been shown that replacing saturated fatty acids with unsaturated fatty acids could lower HDL-C [42]. In our study, DP2 was positively associated with low HDL-C. More future investigations are needed to draw a definite conclusion.

DP3, named the fruit and junk food pattern, was characterized by high consumption of fruits, junk food (including fast foods, ethnic foods and cakes), nuts and eggs. It was significantly associated with an increased likelihood of having MetS, abdominal obesity and high TG. Apparently, the consumption of junk food, which has high energy density and low nutritional value, was an important component of DP3, which could lead to excessive energy accumulation, obesity and other metabolic diseases [43]. Regular consumption of fast foods and sweet snacks could also result in weight gain and insulin resistance [44,45], which play key roles in the development of MetS. Furthermore, some studies had reported that the fructose that naturally exists in fruits was associated with increased TG and BP and was reversely associated with HDL-C [46]. On the contrary, our results showed that DP3 had a decreased risk of low HDL-C, although the *p* for this trend was not statistically significant. We speculated that the bioactive components of fruits and nuts, including fiber and some antioxidants, displayed a beneficial effect on lipid profiles [47,48]. However, the intake of fruits and nuts among Chinese children and adolescents was still relatively low, and this needs to be improved. In addition, the association of egg intake with metabolic syndrome and its components remains controversial so far [24], and more studies are needed to discover the links between certain foods and MetS.

DP4, which referred to the cereals and tubers pattern, was rich in cereals, tubers and vegetables and low in junk food. The consumption of vegetables has been described in the DP1 discussion as a beneficial factor for MetS and its components. Cereals and tubers provide rich dietary fiber, less fat and less protein than animal foods [49]. The fiber from cereals has a strong protective effect against the development of MetS and type 2 diabetes [50]. However, positive relationships were observed between DP4 and MetS, abdominal obesity, high TG and low HDL-C in our study. One explanation is that the refined grains, especially refined wheat in modern society, will result in the great loss of healthy constituents, such as fiber, vitamins, minerals and phytonutrients, which could have beneficial effects on the metabolism of the body [51]. Furthermore, the high carbohydrate content from refined grains would have adverse effects on the profile of metabolic risk [52,53]. Further studies are necessary to explore the effects of the carbohydrate and fiber content in grains on the risks of MetS.

The soy food in DP5 (the beans pattern),which was rich in legumes, mixed beans and tubers, but was low in eggs, is one of the most common dietary components and has been demonstrated to have numerous health benefits. For instance, it is a major food source of isoflavones, phytosterols, lecithin, polyunsaturated fatty acids and dietary fiber [54,55]. We observed that there was a negative association between DP5 and abdominal obesity among children and adolescents aged 7–17. This was consistent with some previous studies [56,57,58]. Although soybean has a long history of tradition as a beneficial food for controlling diabetes and lowering blood glucose [59,60], no relationship between DP5 and FBG was found in the current study. Furthermore, the effect of soybeans on lipid metabolism is still controversial [59,61]. We believe that DP5 was positively correlated with high TG in Chinese children and adolescents, and more studies are needed to verify the relationship between this dietary pattern and lipid metabolism.

The main limitation of this study was the cross-sectional nature of the data, which prevented us from reaching causal conclusions. In addition, the data on food consumption that were collected by the 24-h dietary record and weighing method were not representative of whole-year consumption. Further, the allocation of oil and salt was based on the proportion of the reference for men, which could lead to slight errors in the analysis of individual intake because of individual differences. Fourthly, residual confounding effects cannot be avoided in any observational study. Well-designed, large-scale prospective studies are warranted in the future. In addition to its limitations, our study had strengths as well. Above all, using the scientific sampling method and rigorous quality control measures, the sample size of this study was large and the results can be generalized to the entire population. Furthermore, we controlled for the potential confounders in our analysis as much as possible, including physical activity, family economic condition, smoking and alcohol drinking, which play important roles in disease development.

## 5. Conclusions

Five dietary patterns were obtained through factor analysis among Chinese children and adolescents aged 7–17 in 2016–2017: the animal product and vegetable pattern, condiment pattern, fruit and junk food pattern, cereals and tubers pattern and beans pattern. The cereals and tubers pattern mainly appeared in all subjects, followed by the beans pattern and the fruit and junk food pattern, which demonstrated that there was a predominantly plant-based diet among subjects. The animal product and vegetable pattern may have a positive association with MetS, but was negatively associated with low-HDL-C. The condiment pattern was negatively associated with abdominal obesity and elevated BP, and positively associated with low HDL-C. The fruit and junk food pattern had positive relationships with MetS, abdominal obesity and high TG, but was negatively related to low HDL-C. The cereals and tubers pattern was positively associated with MetS, abdominal obesity, high TG and low HDL-C. The beans pattern was negatively associated with abdominal obesity and positively associated with high TG.

In brief, we identified five mutually exclusive dietary patterns and explored the associations between them and MetS and its components, which may indicate the potential for MetS prevention through effective dietary intervention. Our results could also have benefits in preventing possible type 2 diabetes and cardiovascular disease among Chinese children and adolescents. More large-scale, prospective studies are needed to explore the relationship between dietary patterns and metabolic syndrome.

## Figures and Tables

**Figure 1 nutrients-14-03524-f001:**
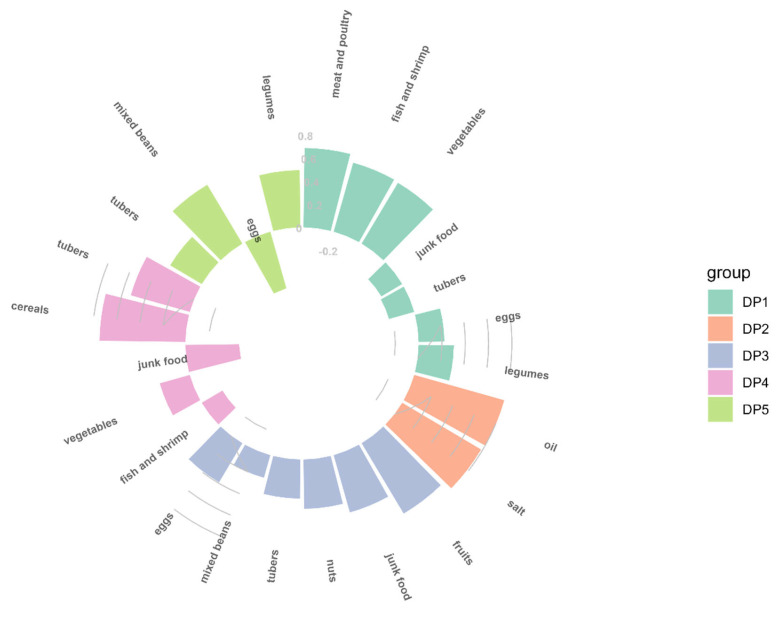
Factor loading of food items in each dietary pattern.

**Table 1 nutrients-14-03524-t001:** Weighted general characteristics [*n* (%)] of children and adolescents in 2016–2017 by MetS group.

Characteristics	Without MetS	With MetS	All
Sex #			
Male	6171 (50.24)	361 (3.14)	6532 (53.38)
Female	6220 (44.40)	319 (2.22)	6539 (46.62)
Residence area *			
Urban	5745 (43.65)	378 (2.93)	6123 (46.58)
Rural	6646 (50.99)	302 (2.42)	6948 (53.42)
Age group *			
Prepubertal	5697 (34.89)	254 (1.53)	5951 (36.42)
Pubertal	3484 (24.66)	216 (1.51)	3700 (26.17)
Post-pubertal	3210 (35.08)	210 (2.32)	3420 (37.41)
Engel’s coefficient			
≥60%	209 (1.53)	11 (0.09)	220 (1.62)
50–59%	282 (2.17)	15 (0.12)	297 (2.29)
40–49%	281 (2.25)	22 (0.18)	303 (2.43)
30–39%	716 (5.38)	40 (0.33)	756 (5.71)
<30%	1844 (14.00)	94 (0.74)	1938 (14.74)
Unknown	9059 (69.32)	498 (3.89)	9557 (73.20)
Physical activity			
0–3 days/week	4317 (33.38)	256 (2.07)	4573 (35.45)
≥4 days/week	5240 (40.50)	275 (2.16)	5515 (42.67)
Unknown	2834 (20.75)	149 (1.12)	2983 (21.88)
Smoking			
Everyday	1375 (10.36)	78 (0.62)	1453 (10.98)
4–6 days/week	525 (4.19)	30 (0.26)	555 (4.44)
1–3 days/week	1674 (13.08)	87 (0.70)	1761 (13.78)
<1 day/week	1831 (14.96)	105 (0.87)	1936 (15.83)
No	6986 (52.06)	380 (2.91)	7366 (54.97)
Alcohol drinking			
Within 30 days	509 (4.95)	27 (0.29)	536 (5.23)
30 days ago	1155 (11.09)	64 (0.63)	1219 (11.71)
Never	10,727 (78.61)	589 (4.44)	11,316 (83.05)
All	12,391 (94.64)	680 (5.36)	13,071

Rao–Scott Chi-square test was applied. # *p* < 0.05; * *p* < 0.0001.

**Table 2 nutrients-14-03524-t002:** Weighted percentages [*n* (%)] of five DPs in children and adolescents aged 7–17 in 2016–2017.

Characteristics	DP1	DP2	DP3	DP4	DP5
Sex *					
Male	1298 (10.35)	1050 (8.59)	1183 (9.28)	1618 (13.33)	1383 (11.82)
Female	1294 (8.89)	1066 (7.73)	1437 (10.11)	1547 (11.05)	1195 (8.84)
Residence area *					
Urban	1678 (12.33)	765 (6.05)	1461 (10.57)	1178 (9.16)	1041 (8.47)
Rural	914 (6.92)	1351 (10.27)	1159 (8.83)	1987 (15.22)	1537 (12.18)
Age group *					
Prepubertal	1259 (7.53)	966 (6.01)	1324 (7.93)	1440 (8.97)	962 (5.98)
Pubertal	761 (5.22)	588 (4.25)	660 (4.51)	890 (6.37)	801 (5.82)
Post-pubertal	572 (6.49)	562 (6.07)	636 (6.95)	835 (9.04)	815 (8.86)
Engel’s coefficient *					
≥60%	53 (0.36)	30 (0.23)	34 (0.23)	44 (0.32)	59 (0.48)
50–59%	54 (0.42)	57 (0.44)	58 (0.44)	64 (0.48)	64 (0.52)
40–49%	60 (0.45)	62 (0.53)	53 (0.43)	63 (0.49)	65 (0.54)
30–39%	149 (1.06)	131 (1.01)	175 (1.25)	156 (1.17)	145 (1.22)
<30%	323 (2.45)	349 (2.69)	406 (2.97)	516 (3.93)	344 (2.70)
Unknown	1953 (14.51)	1487 (11.43)	1894 (14.07)	2322 (17.99)	1901 (15.20)
Physical activity *					
0–3 days/week	938 (7.07)	703 (5.43)	931 (7.01)	1046 (8.19)	955 (7.74)
≥4 days/week	1182 (8.77)	866 (6.92)	1136 (8.47)	1315 (10.30)	1016 (8.22)
Unknown	472 (3.41)	547 (3.97)	553 (3.91)	804 (5.89)	607 (4.70)
Smoking *					
Everyday	311 (2.31)	209 (1.55)	300 (2.20)	349 (2.68)	284 (2.23)
4–6 days/week	93 (0.71)	87 (0.67)	125 (1.01)	133 (1.10)	117 (0.95)
1–3 days/week	335 (2.57)	302 (2.33)	300 (2.25)	412 (3.24)	412 (3.39)
<1 day/week	374 (2.95)	334 (2.81)	396 (3.07)	437 (3.60)	395 (3.38)
No	1479 (10.70)	1184 (8.96)	1499 (10.86)	1834 (13.75)	1370 (10.70)
Alcohol drinking *					
Within 30 days	95 (0.96)	105 (0.98)	96 (0.93)	103 (1.01)	137 (1.35)
30 days ago	241 (2.30)	203 (1.89)	216 (2.11)	257 (2.44)	302 (2.97)
Never	2256 (15.98)	1808 (13.45)	2308 (16.35)	2805 (20.93)	2139 (16.34)
All	2592 (19.24)	2116 (16.32)	2620 (19.39)	3165 (24.38)	2578 (20.66)

Rao–Scott Chi-square test was applied; * *p* < 0.0001.

**Table 3 nutrients-14-03524-t003:** The associations between dietary patterns with MetS and its components by logistic regression.

DP	MetS	Abdominal Obesity	Elevated FBG	Elevated BP	High TG	Low HDL-C
	OR (95%CI)	OR (95%CI)	OR (95%CI)	OR (95%CI)	OR (95%CI)	OR (95%CI)
DP1						
Q1	ref	ref	ref	ref	ref	ref
Q2	1.256 (1.003, 1.572)	1.094 (0.954, 1.254)	1.020 (0.693, 1.502)	1.045 (0.945, 1.156)	0.994 (0.870, 1.136)	0.857 (0.737, 0.997)
Q3	1.225 (0.975, 1.538)	1.036 (0.901, 1.190)	0.827 (0.550, 1.244)	1.020 (0.920, 1.130)	1.023 (0.894, 1.171)	0.668 (0.569, 0.784)
Q4	1.079 (0.851, 1.367)	1.028 (0.893, 1.183)	0.893 (0.592, 1.349)	0.980 (0.882, 1.089)	1.030 (0.897, 1.183)	0.627 (0.530, 0.741)
*p* for trend	0.1546	0.5018	0.5009	0.9304	0.7114	<0.0001
DP2						
Q1	ref	ref	ref	ref	ref	ref
Q2	0.922 (0.744, 1.143)	0.868 (0.761, 0.991)	0.709 (0.481, 1.045)	0.981 (0.887, 1.084)	0.951 (0.833, 1.085)	0.940 (0.801, 1.103)
Q3	0.882 (0.708, 1.097)	0.914 (0.801, 1.043)	0.681 (0.457, 1.013)	0.897 (0.810, 0.992)	0.920 (0.805, 1.051)	0.883 (0.750, 1.039)
Q4	0.920 (0.737, 1.147)	0.819 (0.713, 0.939)	0.902 (0.618, 1.315)	0.946 (0.855, 1.048)	0.933 (0.816, 1.067)	1.172 (1.002, 1.370)
*p* for trend	0.2905	0.0067	0.1350	0.1197	0.2096	0.6832
DP3						
Q1	ref	ref	ref	ref	ref	ref
Q2	1.226 (0.973, 1.543)	1.184 (1.024, 1.368)	1.403 (0.955, 2.063)	1.093 (0.987, 1.211)	1.128 (0.985, 1.292)	0.972 (0.829, 1.140)
Q3	1.191 (0.942, 1.505)	1.345 (1.166, 1.553)	0.840 (0.541, 1.304)	1.026 (0.925, 1.138)	1.126 (0.981, 1.292)	1.054 (0.899, 1.236)
Q4	1.367 (1.079, 1.732)	1.536 (1.329, 1.776)	1.176 (0.769, 1.799)	1.087 (0.977, 1.210)	1.156 (1.003, 1.332)	0.833 (0.700, 0.991)
*p* for trend	0.0159	<0.0001	0.6634	0.1487	0.0287	0.2733
DP4						
Q1	ref	ref	ref	ref	ref	ref
Q2	1.012 (0.805, 1.272)	0.915 (0.795, 1.053)	1.392 (0.945, 2.050)	1.033 (0.933, 1.143)	1.046 (0.914, 1.198)	1.104 (0.935, 1.304)
Q3	1.119 (0.893, 1.403)	1.107 (0.965, 1.271)	0.969 (0.635, 1.480)	1.025 (0.925, 1.135)	1.049 (0.915, 1.202)	1.243 (1.054, 1.465)
Q4	1.315 (1.057, 1.636)	1.357 (1.188, 1.551)	1.157 (0.769, 1.741)	1.100 (0.993, 1.218)	1.165 (1.019, 1.332)	1.371 (1.166, 1.612)
*p* for trend	0.0651	0.0030	0.5030	0.1637	0.0843	0.0005
DP5						
Q1	ref	ref	ref	ref	ref	ref
Q2	0.993 (0.798, 1.235)	0.960 (0.837, 1.100)	1.055 (0.717, 1.553)	0.970 (0.875, 1.075)	1.124 (0.979, 1.290)	0.985 (0.836, 1.160)
Q3	0.889 (0.708, 1.117)	0.995 (0.867, 1.143)	0.965 (0.645, 1.446)	0.910 (0.820, 1.011)	1.182 (1.029, 1.357)	0.942 (0.797, 1.113)
Q4	0.837 (0.669, 1.048)	0.862 (0.751, 0.989)	0.724 (0.472, 1.112)	0.957 (0.864, 1.060)	1.047 (0.912, 1.201)	0.935 (0.792, 1.102)
*p* for trend	0.1725	0.1565	0.3482	0.1839	0.0991	0.4296

The multivariate logistic regression model was adjusted for sex, living area, age, Engel’s coefficient, physical activity, smoking and alcohol drinking. MetS: metabolic syndrome, FBG: fast blood glucose; BP: blood pressure; TG: triglycerides; HDL-C: high-density lipoprotein cholesterol, OR: odds ratio, CI: confidence interval.

## Data Availability

The data are not publicly available according to the policy of the National Institute for Nutrition and Health, Chinese Center for Disease Control and Prevention.

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
