# Peer review of "Association of Dietary Patterns with Metabolic Syndrome in Chinese Children and Adolescents Aged 7–17: The China National Nutrition and Health Surveillance of Children and Lactating Mothers in 2016–2017"

_nutrients, 2022, doi:10.3390/nu14173524_

Round 1
Reviewer 1 Report
A brief summary:
The manuscript is clear, understandable, concise and well written. All parts of the manuscript are described appropriately. The topic is interesting and the text is easy to read. However, there are some remarks which need to be revised before publication.
General concept comments:
· The Abstract is too long. It should include 200 words maximum. Try shortening the existing text and in the description of the results (in the abstract) include only what you found a positive association.
· Suggestion: Introduction could be just one paragraph, without separation, except the last sentence describing the aim of the study.
· Suggestion: In the Keywords separate „children and adolescents“ as two separate keywords
· Suggestion: Always use the same number of decimal places in the article.
· The references are mostly too old. If there is not proper replacement, the authors can include old references (if it is very important for the article), but the literature must mostly include references no older than 5 years. Where possible, please find and replace old literature with newer.
· Please check the text again for some typos.
· Please check English with a professional.
Specific comments:
· In the last sentence of the Introduction, clearly state the aim of the paper, what the research was intended to achieve. Remove the information from where you took the data (“we use the data from the China National Nutrition and Health Surveillance of 69 Children and Lactating Mothers in 2016-2017“), because it is mentioned in the first sentence of Materials and Methods where it should be.
· Line 109 – please make space between words „(TZG)“ and „with“
· Line 179-180: Please rewrite the sentence to make it more clearer.
· Line 204-205: Please rewrite the sentence to make it more clearer.
· Line 206: Instead „participants living in urban“ include „participants living in urban area“.
· Table 3: Correct the Table name in this line and do not use bold in the Table
· Line 242: please exclude the word „indeed“ or replace it with more appropriate word.
· Line 267: As more appropriate term, include „A few studies“ instead „A lot of studies“
· Line 274: A word „related“ or „associated“ is missing in the sentence. Please correct.
· Line 311-313: Please rewrite the sentence to make it more clearer.
· Line 318-319: Please rewrite the sentence.
· Line 347-349 – please include full stop between two sentences.
· Line 357: Instead a term „In a word“ include more appropriate term. Suggestion: „ In a few words“, „In short“, „Finally“, „In conclusion“...or similar
Author Response
Dear Reviewer,
Thank you for your letter and comments concerning our manuscript entitled "Association of dietary patterns with Metabolic syndrome in Chinese children and adolescents aged 7-17: The China National Nutrition and Health Surveillance of Children and Lactating Mothers in 2016-2017"(nutrients-1858652). Based on your suggestions, we have accordingly revised our manuscript. Below you could find the point-to-point response to the questions regarding the manuscript.
We hope that our answers have satisfied your comments and look forward to your response.
Warm regards,

Reviewer 2 Report
Thank you for the opportunity to review this manuscript. The authors have examined the association between dietary patterns and metabolic syndrome and its components. There are several concerns for the authors to consider.
My major concerns:
1. Children and adolescents had different behaviours. I think the authors have enough participants to do the analysis in children and adolescents separately. At least, an interaction or stratified analysis may need to be done.
2. The authors state that 14 groups of foods were included in the dietary pattern analysis. They may need to elucidate what the food items are for each food group.
3. How missing data were treated?
4. Table 1: the authors reported the figures in a wrong way. The percentages in each column for each variable summed should be 100%. For example, summation of the percentages of male and females with MetS should be 100%.
5. Table 2: again, the authors reported the figures in a wrong way. What does “weighted distribution of five DPs” mean? Lines 202-205: the proportion of five DPs was 19.24%, 16.32%, 19.39%, 24.38% and 203 20.66%, respectively. As I know, a continuous score for each pattern would be derived from factor analysis. What’s the proportion? Were participants with a higher score of DP classified as the DP? If so what’s the cut-off point?
6. It seems that the authors only identified several unhealthy dietary patterns that might increase the likelihood of MetS. Why no healthy dietary patterns?
Minor concerns:
1. Lines 184-185: the authors stated that “However, it was noted that the rate of participants who were smoking has been up to 45.03%”. It is hardly believable that the prevalence of smoking is so high in children and adolescents. The authors may need to check it out.
2. Line 192: there are three decimal places for these percentages but two decimal places for other percentages. It should be consistent.
3. The authors stated that the Animal products and vegetables pattern had a positive association with MetS. However, Table 3 shows that individuals in quartile 2 but not quartiles 3 and 4 had a higher likelihood of MetS. So, this is not a linear relationship. The authors need to be cautious about this.
4. There are many grammar errors. The authors need to check throughout the text.
Author Response
Dear Reviewer,
Thank you for your letter and comments concerning our manuscript entitled "Association of dietary patterns with Metabolic syndrome in Chinese children and adolescents aged 7-17: The China National Nutrition and Health Surveillance of Children and Lactating Mothers in 2016-2017"(nutrients-1858652). Based on your suggestions, we have accordingly revised our manuscript. Below you could find the point-to-point response and explanation to the questions regarding the manuscript.
We hope that our answers have satisfied your comments and look forward to your response.
Warm regards,
